# *Arenga pinnata* Resistant Starch Modulate Gut Microbiota and Ameliorate Intestinal Inflammation in Aged Mice

**DOI:** 10.3390/nu14193931

**Published:** 2022-09-22

**Authors:** Minhong Ren, Meng-Yun Li, Lin-Qian Lu, Yuan-Sen Liu, Feng-Kun An, Kai Huang, Zhen Fu

**Affiliations:** 1Guangxi Vocational & Technical Institute of Industry, Nanning 530001, China; 2College of Light Industry and Food Engineering, Guangxi University, Nanning 530004, China; 3The Center of Nanning New Technology Entrepreneur, Nanning 530007, China

**Keywords:** *Arenga pinnata*, resistant starch, gut microbiota

## Abstract

This study aimed to compare the regulatory effects of *Arenga pinnata* retrograded starch (APRS), *Arenga pinnata* starch (APS), and whole *Arenga pinnata* flour (APF) on gut microbiota and improvement of intestinal inflammation in aged mice. APF, APS, and APRS altered gut microbiota composition and exhibited different prebiotic effects. *Bifidobacterium* showed the greatest increase in feces of aged mice fed APF. The abundance of genus *Lachnospiraceae*_NK4A136 was highest in the APS group. APRS supplementation led to a greatest increasement in abundance of *Lactobacillus*, *Roseburia*, and *Faecalibacterium prausnitzii*. APRS induced significantly more short-chain fatty acid (SCFAs) production than APF and APS. APF, APS, and APRS treatments improved intestinal inflammation in aged mice and the order of ameliorative effect was APRS > APS > APF. APRS significantly decreased relative mRNA expression of pro-inflammatory cytokines (IL-6, IL-1β, and TNF-α) and increased anti-inflammatory cytokines (IL-10). In addition, APF, APS, and APRS significantly downregulated the relative mRNA expression of senescence-associated gene p53 and upregulated the expression of anti-aging gene Sirt1. These results provide potentially useful information about the beneficial effects of *Arenga pinnata* products on human health.

## 1. Introduction

The gut microbiome, composed of approximately 100 trillion microbes [1], is an interactive ecosystem that plays an important role in the host’s health [2,3]. The characteristics of aging include a gradual decline in the body’s health and deterioration of physiological functions. Aging is associated with changes in the gut microbiota that might result in dysbiosis and illnesses [4]. It is reported that specific beneficial bacteria, such as Bifidobacterium, are depleted in elderly individuals [4], and the abundance of Lactobacillus in the elderly from a long-lived village was higher than in controls [5]. Therefore, regulating the gut microbiota may provide an opportunity to improve health among the elderly [6].

Dietary interventions, especially dietary fiber, could modify the activity and composition of the intestinal microbiota [6]. Diets supplemented with resistant starch (RS) lead to alterations in the composition and function of the gut microbiota in a healthy host [7]. RS is perceived as a multifunctional regulator for treating metabolic dysfunction, by modulating gut microbiota, gut peptides, circulating growth factors, and inflammatory mediators [8].

RS is resistant to enzymatic digestion in the stomach and small intestine but could be fermented by gut microbiota in the large bowel [9]. In general, RS is subdivided into five categories, including physically entrapped starch (RS1), nongelatinized native starch granule (RS2), retrograded starch (RS3), chemically modified starch (RS4), and amylose−lipid complexes (RS5) [9,10]. RS plays a pivotal role in providing metabolic and colonic health benefits and has superior functional properties as a form of dietary fiber, including good sensory qualities [11] and increasing the production of short-chain fatty acids (SCFAs) [12].

Accumulating evidence has indicated that the consumption of RS prepared from cereals, tubers, and legumes plays important roles in promoting several health benefits [13,14]. Different RS types had different influences on the gut microbiota composition. The functionality and health benefits of RS have been widely studied in in vitro and in vivo experiments; however, differences in function and performance between different types of RS are unclear, especially in an aged host.

*Arenga pinnata* is a multipurpose palm species from which a variety of foods and beverages, timber commodities, biofibers, biopolymers, and biocomposites have been produced [15]. As an important source of starch in subtropical and tropical areas, *Arenga pinnata* flour (APF) is generally prepared from the pulp of AP [16]. APS has high amylose content and is an ideal preparation material for RS [17]. Meanwhile, preliminary experiments of RS3 from APS through in vitro fermentation suggested that *Arenga pinnata* retrograded starch (APRS) had prebiotic effects. Therefore, the objective of this study was to investigate the regulatory effects of *Arenga pinnata* retrograded starch (APRS containing RS3), *Arenga pinnata* starch (APS containing RS2), and whole *Arenga pinnata* flour (APF mainly containing RS1 and RS2) on gut microbiota and explore the correlations between the key phylotypes of gut microbiota and the metabolic pathway in aged mice. The results of this study could provide applications to support the development of functional, healthy, and fiber-rich dietary *Arenga pinnata* products.

## 2. Materials and Methods

### 2.1. Materials

*Arenga pinnata* flour (APF) (main components: starch 62.4%, protein 4.7%, fat 0.4%, fiber 5.4%, and water 13.2%; RS content is 50.8%) was prepared from the pulp of sugar palm (*A. pinnata* (*Wurmb.*) *Merr*) cultivated in Longzhou county, Guangxi province, China, and APS (main components: starch 87.8%, fat 0.2%, protein 1.6%, fiber 0.9%, and water 9.2%; RS content is 51.2%) was isolated from the APF [16,18]. APRS (RS content is 51.8%) was prepared using a published method [17]. Basal feed for mice (in line with the Chinese standard, “Feed Health Standard” (GB13078-2017), was produced by Beijing Keao Xieli Feed Co., Ltd. (Beijing, China).

### 2.2. Animal Experiments

The control group (CK) was fed a basal diet; APF, APS, or APRS groups were fed basal diets with 10% APF, APS, or APRS, respectively. The protocols and experiments were approved by the Ethics Committee of Guangxi University. Twenty-four C57BL/6J naturally aged mice (SPF, male, 18 months old) were purchased from the Experimental Animal Center of Guangxi Medical University (Production license no. L20160258SCXK Gui 2020-0003).

Mice were housed under a condition of 24 ± 2 °C, relative humidity of 55 ± 5%, and a 12 h light/dark cycle. Mice were housed in a standard cage (6 mice per cage). All efforts were made to minimize animal suffering. After a week of acclimatization feeding, mice were randomly divided into four groups (with 6 mice per group): a control group (CK), APF group, APS group, and APRS group. Mice in all groups had free access to feed and water.

Animal experiments lasted for 8 weeks. The general health of the animals was checked daily, and body weight and food intake were recorded weekly. At the end of the experiment, fresh feces samples were collected in sterilized Eppendorf (EP) tubes and immediately preserved at −80 °C for further analysis. Then the mice were sacrificed (killed by neck amputation after ether anesthesia). The small intestine tissues were excised carefully and quickly. The jejunum segment of the small intestine was precisely dissected. The middle segment of jejunum tissues (about 1 cm) was excised and fixed in 10% formalin for more than 24 h for histological analysis. Meanwhile, the rest of the jejunum was collected in aseptic EP tubes and immediately stored at −80 °C for subsequent analysis.

### 2.3. Histopathological Analysis

The jejunum tissues were processed by dehydration, cleaning, and infiltration. Then, the samples were embedded in paraffin wax and sectioned with a paraffin slicer. The sectioned samples were stained with hematoxylin and eosin (H&E) and observed under an inverted microscope. Images were captured at 100× magnification for jejunum tissues.

### 2.4. SCFA Analysis

The concentrations of SCFAs including acetate, propionate, and butyrate in the feces were analyzed using an external standard method. Briefly, 0.2 g of feces were suspended in 2 mL of saturated NaCl solution. After vortexing uniformly for 30 min, the mixtures were centrifuged at 12 000× *g* for 10 min. The supernatant was acidified with 100 μL of 80% H_3_PO_4_, and then extracted with 2 mL of ethyl ether for 10 min. The concentrations of SCFAs were determined using an 8890N gas chromatograph equipped with a FID detector and HP-INNOwax capillary column (30 m × 0.25 mm × 0.25 µm film thickness) (Agilent Technologies Inc., Santa Clara, CA, USA). Separation was achieved using the following conditions: temperatures of the injector and detector were both 250 °C; flow rate of the carrier gas (high purity nitrogen) was kept at 1.5 mL/min; injected quantity was 1 µL and split ratio was 10:1; initial column temperature was 100 °C, increased to 150 °C at a rate of 8 °C/min, ramped up to 170 °C at 5 °C/min, and then finally raised to 230 °C at the rate of 30 °C/min and kept at 230 °C for 2 min.

### 2.5. Microbial Genomic DNA Extraction and 16S rRNA Gene-Based Illumina MiSeq Sequencing and Date Analysis of Feces

The methods used for fecal microbial genomic DNA extraction, library construction, sequencing, and bioinformatics analysis were the same as we previously described [5]. The analysis of α-diversity, which included the calculation of observed OTU numbers, Chao1, Shannon index, and Simpson index in all samples, was estimated by MOTHUR (v1.31.2) at the OTU level. The Good’s coverage was selected to characterize the sequencing depth, and Chao1 was used to identify community richness, and the Shannon and Simpson indices reflected community diversity. Principal coordinate analysis (PCoA) was performed by QIIME (v1.8.0). To identify the statistical differences between groups, we performed linear discriminant analysis (LDA) effect size (LEfSe) analysis with an LDA score threshold of 4. The Venn plots in OTUs were plotted using the R package “VennDiagram”, version 3.1.1.

### 2.6. Quantitative Real-Time PCR Analysis of the Feces

For real-time qPCR, microbial genomic DNA was extracted from the feces by using a stool genomic DNA extraction kit (Solarbio, Beijing, China), according to the manufacturer’s recommendations. The concentration of DNA was measured using an Infinite M200 PRO continuous wavelength multifunctional microporous detector (Tecan, Männedorf, Switzerland). Total intestinal bacteria and four genera were quantified from each fecal DNA sample using real-time PCR. DNA Amplification and detection was performed using the Roche LightCycler 96 real-time PCR instrument (Roche Diagnostics Co., Ltd., Basel, Switzerland). Sample analysis was performed in a total volume of 20 μL using SYBR Green qPCR Master Mix. Each reaction consisted of 1 µL of template DNA (containing 30 ng DNA), 1 µL of forward primer and reverse primer with a concentration of 10 µM, 7 µL of ddH_2_O, and 10.0 µL of 2× Realtime PCR Super Mix (Mei5 Biotechnology Co., Ltd., Beijing, China). Primer sequences of the target genus are listed in Table 1. Real-time PCR conditions included an initial denaturation step at 95 °C for 5 min and an amplification step, followed by 40 cycles of denaturation at 95 °C for 30 s, annealing at the optimum annealing temperature of primers (Table 1) for 30 s, elongation at 72 °C for 1 min, and then re-elongation at 72 °C for 8 min. At the end of the PCR assay, a dissociation curve analysis was performed to examine non-specific products.

### 2.7. Total RNA Isolation and Quantitative Real-Time PCR Analysis

Total jejunum tissue RNA was extracted using a total RNA extraction kit (Solarbio, Beijing, China), according to the manufacturer’s instructions, and concentration and purity were determined using an Infinite M200 PRO continuous wavelength multifunctional microporous detector (Tecan, Männedorf, Switzerland). The cDNA was synthesized using a reverse transcriptase kit (Beyotime, Shanghai, China), following the instructions provided by the manufacturer. Real-time PCR was performed using the SYBR Green Realtime PCR Master Mix (Mei5 Biotechnology Co., Ltd., Beijing, China) in a Roche LightCycler 96 real-time PCR instrument (Roche Diagnostics Co., Ltd., Basel, Switzerland). Primer sequences of all target genes are listed in Table 2.

Each reaction contained 1 µL of template cDNA, 1 µL of forward primer and reverse primer with a concentration of 10 µM, 7 µL of ddH_2_O, and 10.0 µL of 2× Realtime PCR Super Mix (Mei5 Biotechnology Co., Ltd., Beijing, China). Real-time PCR conditions were carried out as previously described [20]. At the end of the PCR analysis, dissociation curve analysis was performed to check for nonspecific products. All genes were compared with the housekeeping control gene β-actin using the 2^−∆∆Ct^ calculation method.

### 2.8. Statistical Analysis

All data were expressed as means ± standard deviations. Statistical analyses were performed with SPSS V22.0 statistical software for Windows (SPSS Inc., Chicago, IL, USA) to determine the statistical differences among groups using parametric methods (analysis of variance, ANOVA) and nonparametric statistical methods (Mann–Whitney U-test and Kruskal–Wallis test). The differences among more than two groups were analyzed using a one-way ANOVA followed by a LSD test. Significance was set at *p* < 0.05.

## 3. Results

### 3.1. Microbial Diversity Analysis

To investigate the influences of APF, APS, and APRS on the gut microbiome, a total of 746,093 high-quality 16S rRNA sequence reads from the fecal samples of mice were analyzed after 8 weeks of feeding with APF, APS, and APRS, respectively. The average sequence reads for all the samples were 53,139 ± 1943. Rarefaction curve and OTU rank curve analysis were performed, and the sequencing depth of this study was found to be adequate with a Good’s coverage of 0.9992 ± 0.0001.

The α-diversity index was used to calculate the species diversity within the microbial community. Richness and evenness were two principal factors influencing α-diversity. As shown in Figure 1, supplementation with APF and APS significantly decreased α-diversity levels, as evaluated by observed OTU number (sobs), Chao1 index, and Shannon and Simpson indices compared with the control group (Figure 1A–D). The Venn diagram (Figure 1E) shows that there were 273 OTUs common to all groups. There were 16 unique OTUs for control, 10 for APF, 9 for APS, and 33 for APRS. In addition, the principal coordinate analysis (PCoA) (Figure 1F), based on the unweighted UniFrac distance showed that the gut microbial communities were clustered and separated among groups, suggesting that APF, APS, and APRS had different effects on the gut microbiota.

### 3.2. Microbial Composition Analysis

At the phylum level (Figure 2A), a total of 10 bacterial phyla (Firmicutes, Proteobacteria, Desulfobacterota, Bacteroidetes, Actinobacteria, Patescibacteria, Campilobacterota, Deferribacterota, Cyanobacteria, and Fusobacteriota) were detected. Among them, the relative abundances of the six phyla, Firmicutes, Proteobacteria, Desulfobacterota, Bacteroidetes, Actinobacteria, and Patescibacteria, accounted for 99%. APF induced a significant reduction in Firmicutes abundance (*p* < 0.01) and increases in Bacteroidetes (*p* < 0.05) and Actinobacteria abundances (*p* < 0.001). The relative abundances of Bacteroidetes and Actinobacteria in the APS group were also increased compared with the control group (9.91 ± 2.45 vs. 3.12 ± 0.55% and 9.06 ± 3.46 vs. 1.15 ± 0.40%, respectively). In addition, the relative abundances of Proteobacteria in each group were reduced relative to the control group.

At the family level (Figure 2B), APF supplementation significantly increased the relative abundances of *Bifidobacteriaceae*, *Muribaculaceae*, and *Prevotellaceae*, and significantly decreased the abundance of *Lachnospiraceae* (*p* < 0.05) compared with the control group (CK group). APS also increased the abundances of *Bifidobacteriaceae* and *Muribaculaceae*. APRS treatment led to significant increases in the abundances of *Ruminococcaceae*, *Monoglobaceae*, and *Saccharimonadaceae* (*p* < 0.05). The relative abundance of *Sutterellaceae* decreased in the three treatment groups compared with the CK group.

There were significant differences in the gut microbiota composition at the genus level (Figure 2C). The relative abundance of *Prevotellaceae*_UCG-001 (*p* < 0.05) and Bifidobacterium (*p* < 0.01) was significantly increased in APF groups than in the other three groups. The relative abundance of Lachnospiraceae_NK4A136, a potential probiotic [21], was significantly higher in the APS group than in the other groups (*p* < 0.01), whereas APRS supplement increased the relative abundances of *Ruminococcaceae*_*UCG*-014, *Eubacterium*_*xylanophilum*, *Roseburia*, *Monoglobus*, and *Butyricicoccus*. Moreover, the relative abundance of *Lactobacillus* in APRS group was increased compared with the control group.

The LDA score, generated from the linear discriminant analysis effect size analysis, showed distinct gut microbiota compositions among mice from all groups. To identify specific bacteria that were significantly changed among groups, the LEfSe analysis with an LDA score log10 > 4 was employed. Based on the comparison of the four groups, a histogram of LDA scores was plotted and 4, 8, 3, and 10 discriminating biomarkers in the microbiota of the control, APF, APS, and APRS groups were identified, respectively (Figure 3). A variety of beneficial bacteria such as *Bifidobacterium*, *Lachnospiraceae*_*NK4A136*, and *Ruminococcaceae*_*UCG*-014 were significantly enriched in APF, APS, and APRS groups, respectively.

In order to further confirm the abundances of probiotics *Bifidobacterium* and *Lactobacillus* and SCFA-producing bacteria *Roseburia* and *Faecalibacterium prausnitzii* in mice feces, real-time quantitative PCR was used to detect the relative contents of these four bacteria. As shown in Figure 4, the contents of *Bifidobacterium*, *Lactobacillus*, *Roseburia*, and *Faecalibacterium prausnitzii* were increased by APF, APS, and APRS dietary interventions. APF had the most effective upregulation on *Bifidobacterium* content, followed by APS and APRS, which was consistent with the results of the high-throughput sequencing. The contents of *Lactobacillus*, *Roseburia*, and *Faecalibacterium prausnitzii* were significantly improved by APRS, which was consistent with the changes in SCFAs in the following section.

### 3.3. Changes in SCFAs

The content of SCFAs in fecal samples were measured and are shown in Figure 5. The concentration of acetate, propionate, and butyrate in APF, APS, and APRS groups were higher than in the control group. The APRS group had the greatest content of acetate, propionate, and butyrate. These results suggest that APF, APS, and APRS, containing different RS, have different effects on SCFA production. In addition, the abundance of SCFA-producing bacteria *Roseburia* and *Faecalibacterium prausnitzii* significantly increased by APRS intervention.

### 3.4. Effects of AP Resistant Starch on Jejunum Histology in Aged Mice

The jejunum tissues were stained with H&E to investigate morphological changes (Figure 6). Representative pictures show that the length of the jejunum villus obviously increased, whereas inflammatory infiltration decreased, in the three treatment groups, compared with the control group.

### 3.5. Effects of AP Resistant Starch on mRNA Gene Expression in Jejunum of Aged Mice

#### 3.5.1. G-Protein Receptors (GPR) of SCFAs

SCFAs act as endogenous ligands for G-protein-coupled receptors (GPCRs), activating their effects in the organism. The best-studied SCFA receptors are GPR41 and GPR43, which were later renamed free fatty-acid receptor 3 (FFAR3) and FFAR2, respectively [22]. The mRNA expression of GPR41 and GPR43 in the jejunum was detected (Figure 7). Compared with the CK group, the mRNA expression of GPR41 was significantly upregulated, 1.5-fold in the APF group, 4-fold in the APS group, and 7-fold in the APRS group; the mRNA expression of GPR43 was upregulated about 2-fold, 4-fold, and 10-fold in APF, APS, and APRS groups, respectively.

#### 3.5.2. Inflammatory Markers and MAPK Signaling Pathways

To evaluate the influences of APF, APS, and APRS on inflammatory levels in aged mice, we examined the concentrations of proinflammatory cytokine tumor necrosis factor-α (TNF-α), interleukin-6 (IL-6), and interleukin-1β (IL-1β), and anti-inflammatory cytokine interleukin-10 (IL-10) in jejunum tissue (Figure 8). Administrations of APF, APS, and APRS significantly decreased relative mRNA gene expression of TNF-α, IL-6, and IL-1β, and increased IL-10 mRNA expression levels, compared with the CK group (*p* < 0.05). These results indicate that APF, APS, and APRS treatments could improve the level of anti-inflammatory factors and reduce the level of proinflammatory factors in aged mice. The order of ameliorative effects with respect to inflammation was APRS > APS > APF.

The MAPK (mitogen-activated protein kinase) pathway has four main branching routes: ERK (extracellular signal-regulated kinase), JNK (C-Jun amino terminal kinase), P38/MAPK (P38 mitogen-activated protein kinase), and ERK5. Among them, ERK is mainly responsible for regulating cell growth and differentiation, JNK and P38/MAPK signaling pathways play important roles in stress responses, such as inflammation and apoptosis. As can be seen from Figure 8, APF, APS, and APRS significantly downregulated the relative expression levels of p38 and JNK1 in the jejunum tissues of aged mice, suggesting that AP resistant starch can downregulate inflammatory signaling pathways. APRS was the most effective, followed by APS, and finally APF.

#### 3.5.3. Aging-Related Gene Expression

In addition, the relative mRNA expressions of senescence-associated genes were investigated (Figure 9). The results showed that APF, APS, and APRS could significantly downregulate the relative expression of the pro-aging gene p53 of aging mice (*p* < 0.01) and significantly upregulate the relative expression of anti-aging gene Sirt1 (*p* < 0.01). APRS had the strongest effect, followed by APS and APF.

## 4. Discussion

Accumulating research has found that RS has beneficial influences on gut microbiota and SCFA production [23,24]. RS with different characteristics induced different changes to the gut microbiota. Previous studies have reported that RS1, RS2, RS3, RS4, and RS5 could reduce the α-diversity of microbial communities [10,25,26,27] and their respective reductive effects on α-diversity were different. Liang et al. also reported that potato RS2 and RS4 were more effective in reducing the α-diversity of microbial communities [27]. In this study, APS, mainly containing RS2, showed a greater inhibitory effect on the α-diversity of microbial communities. The differences may be attributable to the physical and chemical properties of RS [8,28].

It was reported that mice fed with raw potato starch or RS2 showed a lower abundance of *Proteobacteria* [29,30], which is a major phylum that includes lipopolysaccharide-producing species and is related to dysbiosis and the progression of several diseases [30,31]. Qin et al. [10] stated that RS3 and RS5 shaped the microbial community and promoted the proliferation of specific bacteria. The probiotic effects and healthy benefits of RS have been demonstrated in both in vivo and in vitro studies but have rarely been demonstrated in an aging model. The results of our study suggest that APF, APS, and APRS, containing different RS, also have different influences on the gut microbiota in aged mice. APF, as a whole plant food, contains a higher fiber content and polyphenols, which could facilitate the growth of *Bifidobacteria* and *Lactobacilli*, which are considered beneficial to human health [32,33,34]. It has been reported that the elderly experience depletions of certain beneficial bacteria, such as *Bifidobacterium*, which can reduce gut permeability, cause intestinal inflammation, and produce SCFAs [35]. APF dietary interventions appear the most effective in upregulating *Bifidobacterium* abundance. However, the APF intervention showed a lower growth in SCFA production (acetate, propionate, and butyrate), which are key microbial metabolites in the gut and exert multiple beneficial effects on the host [36,37,38,39,40].

The different molecular structures of RS significantly influenced SCFA production, although the RS came from the same source [10,27]. In addition, changes in SCFAs may be related to alterations in the abundances of specific bacterial groups after dietary intervention [41]. It has been reported that *Bifidobacterium* significantly improved acetate production, but did not contribute to the production of propionate and butyrate [42]. *Lactobacillus* was negatively correlated to the production of propionate and butyrate [27]; however, the growth of *Prevotella* could increase SCFAs such as acetate, propionate, and butyrate production [43]. In this study, the relative content of *Roseburia* and *Faecalibacterium prausnitzii*, which are typical SCFA-producing bacteria, was highest following APRS intervention. The increase in SCFA production resulted in greater expression of SCFA receptors (GPR41 and GPR43) in intestinal epithelial cells.

APRS and APS intervention induced more SCFA production, particularly butyrate, which can control the expression of inflammatory cytokines, such as IL-10, IL-1β, IL-6, and TNF-α, by epithelial cells [44,45]. The reduction in TNF-α, IL-6, and IL-1β levels and increase in IL-10 levels may have been due to activation of the JNK and p38 MAPK signaling pathways by SCFAs activating GPR41 and GPR43 in epithelial cells [22,46]. Additionally, SCFAs such as butyrate may have indirect effects on inflammation by maintaining a robust epithelial barrier [47].

It is well-known that the pro-inflammatory response shows a progressive increase with an increase in age [48,49]. During inflammatory aging, the balance of the gut microbiota is gradually lost, and the composition and diversity of the gut microbiota are altered, causing chronic systemic low-level inflammation [50,51,52] and resulting in the deterioration of intestinal tissue function and an intensified inflammatory response of the body [53]. After dietary intervention, APF, APS, and APRS significantly downregulated the relative mRNA expression of the p53 aging gene in aging mice and significantly upregulated expression of anti-aging gene SIRT1, suggesting that RS dietary intervention had potential anti-aging effects and APRS had the best effect, followed by APS and APF. In short, *Arenga pinnata* resistant starch can improve the intestinal microflora structure of aging mice, increase production of SCFAs, and further activate GPR41 and GPR43 in intestinal tissue to mediate p38 and JNK1 MAPK inflammatory signaling pathways. These outcomes reduce intestinal inflammation and improve intestinal barrier function to achieve anti-aging effects. APRS significantly upregulated relative content of SCFA-producing bacteria and resulted in the highest SCFA content. Therefore, APRS most effectively reduced intestinal inflammation and improved intestinal barrier function to achieve anti-aging effects.

## 5. Conclusions

APF, APS, and APRS could regulate the gut microbiota composition and exhibited different prebiotic effects in aged mice. APRS and APS significantly increased acetate, propionate, and butyrate levels in the feces of aged mice. APRS extremely significantly upregulated SCFA-producing bacteria and resulted in the greatest SCFA content. *Arenga pinnata* resistant starch can improve the intestinal microflora structure of aging mice, increase the production of SCFAs, and further activate GPR41 and GPR43 in intestinal tissue to mediate p38 and JNK1 MAPK inflammatory signaling pathway to reduce intestinal inflammation and improve intestinal barrier function. APRS was most effective in reducing intestinal inflammation and improving intestinal barrier function to achieve anti-aging effects. These results provide useful information about the beneficial effects of *Arenga pinnata* products on human health.

## Figures and Tables

**Figure 1 nutrients-14-03931-f001:**
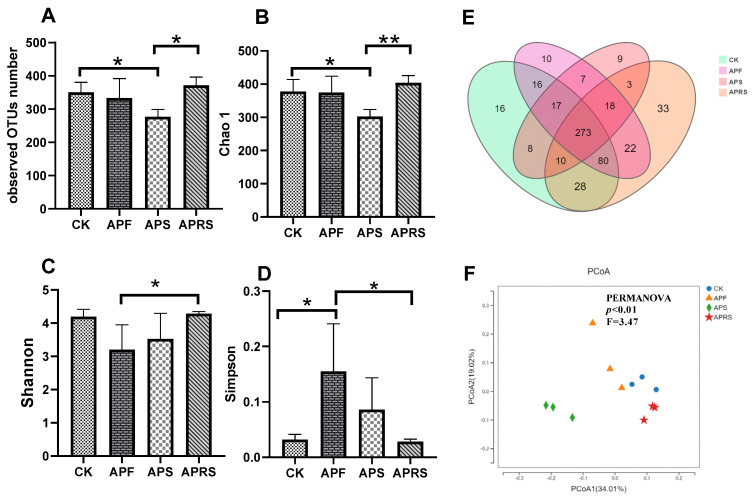
Alpha- and beta-diversity comparisons of fecal microbiome from mice after 8 weeks of feeding with APF, APS, and APRS. (**A**) Observed OTU number, (**B**) Chao 1, (**C**) Shannon, and (**D**) Simpson, * *p* < 0.05, ** *p* < 0.01 (one-way ANOVA). (**E**) Venn diagram showing the unique and shared OTUs among the groups. (**F**) PCoA plot of beta diversity based on unweighted UniFrac distance.

**Figure 2 nutrients-14-03931-f002:**
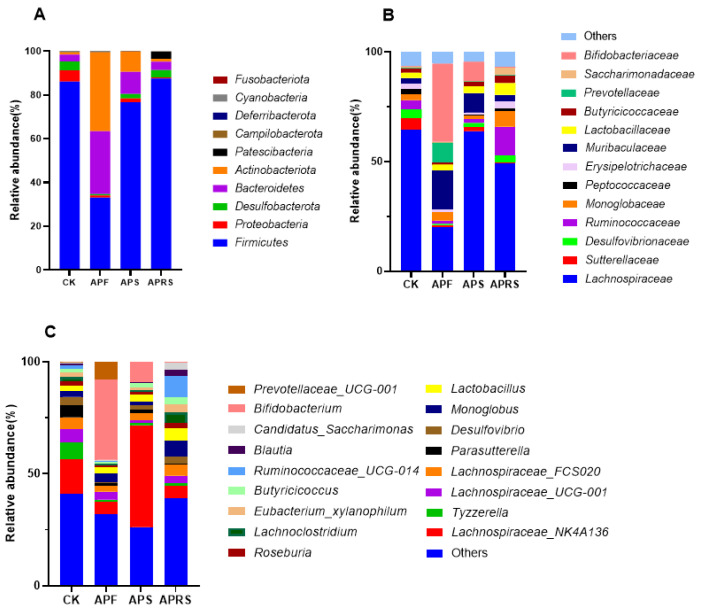
Fecal microbiota composition from mice after 8 weeks of feeding with APF, APS, and APRS. The composition of gut microbiota at the (**A**) phylum, (**B**) family, and (**C**) genus level.

**Figure 3 nutrients-14-03931-f003:**
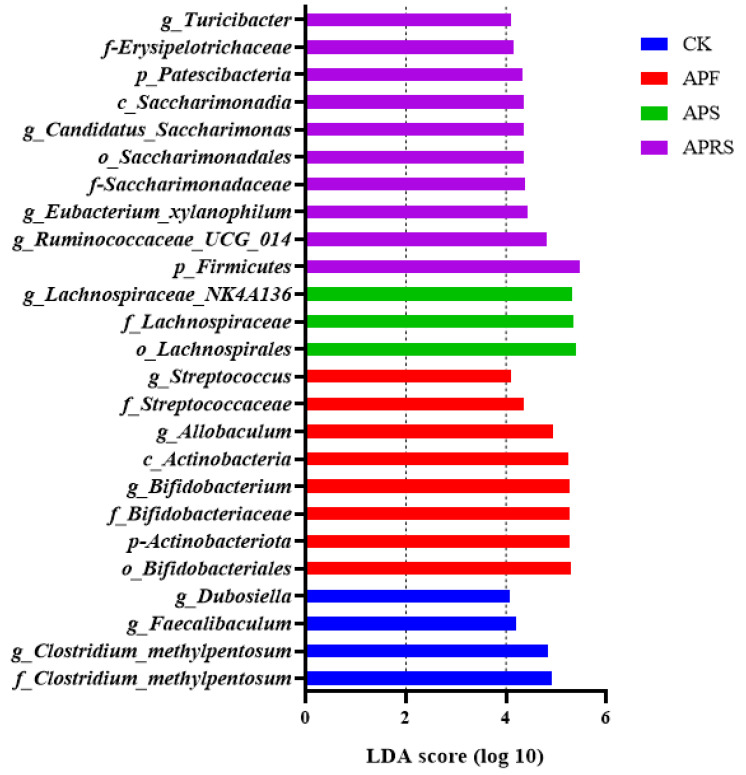
Linear discriminant analysis effect size (LEfSe) showing the contribution of different bacteria to the differences among the groups.

**Figure 4 nutrients-14-03931-f004:**
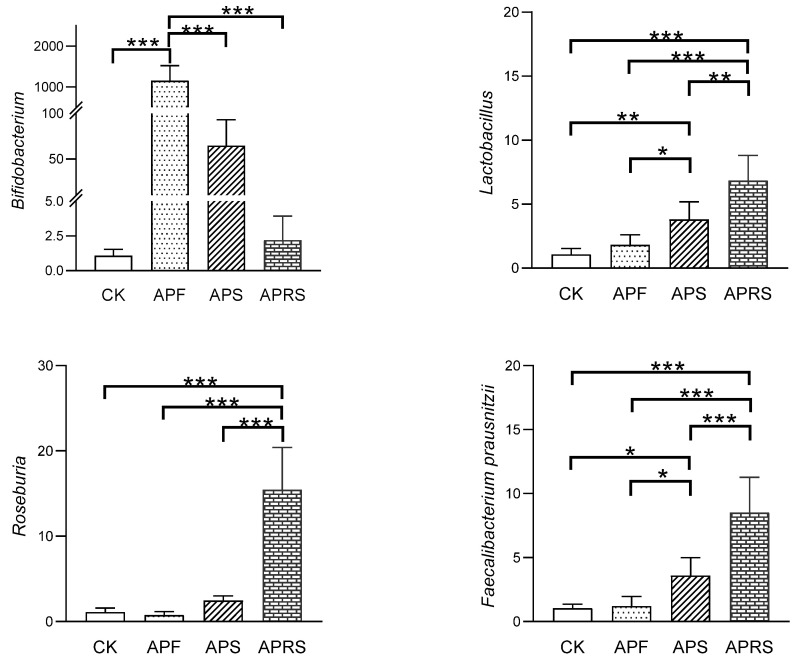
Changes in the relative abundances of *Bifidobacterium*, *Lactobacillus*, *Roseburia*, and *Faecalibacterium p**rausnitzii* under dietary intervention (* *p* < 0.05, ** *p* < 0.01, *** *p* < 0.001, one-way ANOVA with LSD test).

**Figure 5 nutrients-14-03931-f005:**
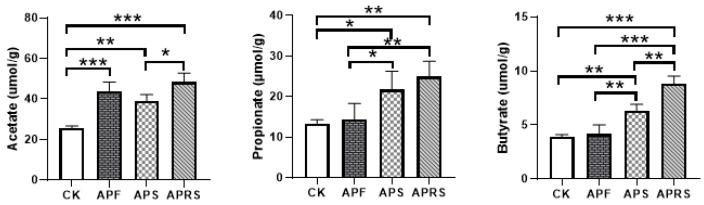
Concentration differences of acetate, propionate, and butyrate in fecal samples (* *p* < 0.05, ** *p* < 0.01, *** *p* < 0.001, one-way ANOVA with LSD test).

**Figure 6 nutrients-14-03931-f006:**
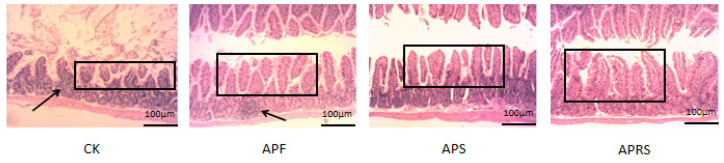
The H&E staining of the jejunum of aged mice after intervention in different groups (magnification, 100×). The black arrows show inflammatory cell infiltration; the black squares show the length of the jejunum villus.

**Figure 7 nutrients-14-03931-f007:**
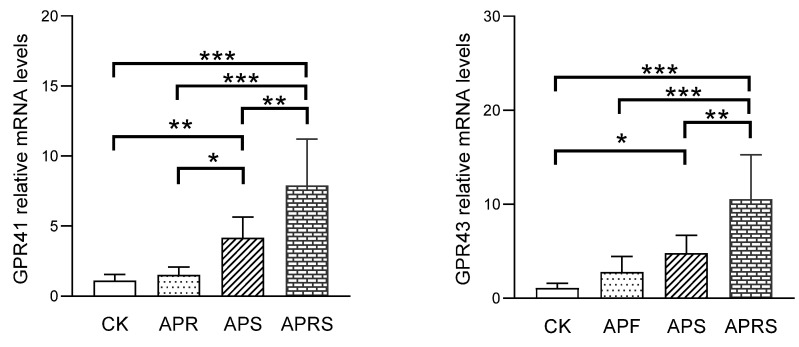
The relative mRNA expression of SCFA receptors in jejunum tissue of aged mice (* *p* < 0.05, ** *p* < 0.01, *** *p* < 0.001, one-way ANOVA with LSD test).

**Figure 8 nutrients-14-03931-f008:**
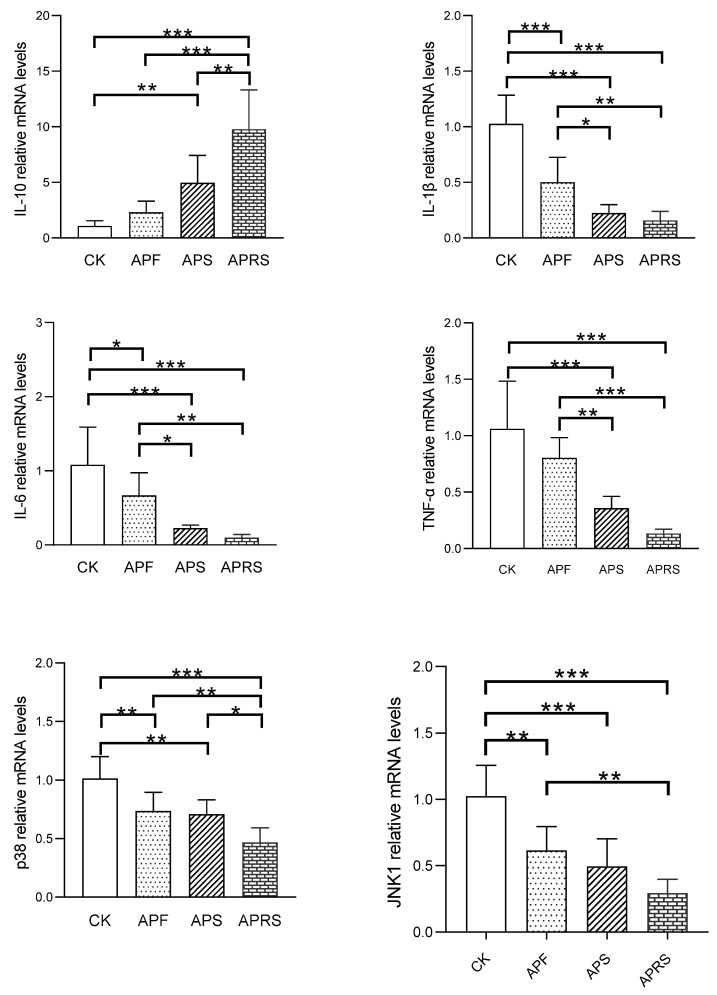
Effects of AP on the relative mRNA expression of inflammatory cytokines and genes related to MAPK signaling pathways in jejunum tissue of aged mice (* *p* < 0.05, ** *p* < 0.01, *** *p* < 0.001, one-way ANOVA with LSD test).

**Figure 9 nutrients-14-03931-f009:**
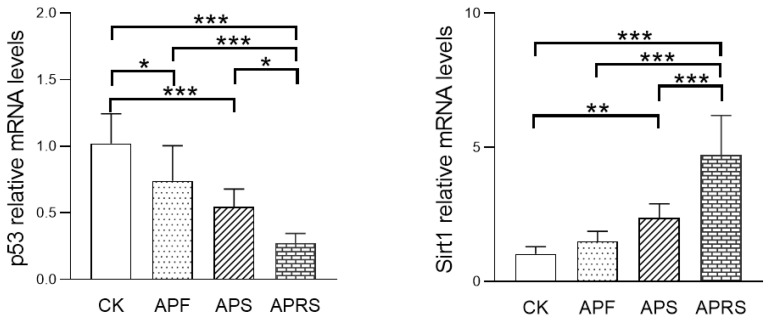
Effects of AP on the relative mRNA expression of senescence-associated genes in jejunum tissue of aged mice (* *p* < 0.05, ** *p* < 0.01, *** *p* < 0.001, one-way ANOVA with LSD test).

**Table 1 nutrients-14-03931-t001:** Genus-specific primers used for real-time qPCR.

Genus	Forward (F) and Reversed (R) Primer Sequence (5′–3′)	Annealing Temperature (°C)	Product Size (bp)	Reference
Total intestinal bacteria	F: ACTCCTACGGGAGGCAGCAG R: ATTACCGCGGCTGCTGG-3′	60	146	
Lactobacillus	F: AGCAGTAGGGAATCTTCCA R: CACCGCTACACATGGAG	58	341	
Roseburia	F: GCGGTRCGGCAAGTCTGA R: CCTCCGACACTCTAGTMCGAC	60	81	[19]
Faecalibacterium	F: GGAGGAAGAAGGTCTTCGG R: AATTCCGCCTACCTCTGCACT	60	248	[19]
Bifidobacterium	F: GGGTGGTAATGCCGGATG R: CCACCGTTACACCGGGAA	60	243	

**Table 2 nutrients-14-03931-t002:** Gene-specific primers used for real-time qPCR.

Gene Name	GenBank Access No.	Forward (F) and Reversed (R) Primer Sequence (5′–3′)	Product Length (bp)
Sirt1	NM_001159589.2	F: CCAGACCCTCAAGCCATGTT R: TTGGATTCCTGCAACCTGCT	201
p53	NM_001127233.1	F: GTGCTCACCCTGGCTAAAGT R: AGGAGGATGAGGGCCTGAAT	107
GPR41	NM_001033316.2	F: CGGCTCACTGTAGTGTGGTT R: AGTCGTACAGGCAGGAGGAT	127
GPR43	NM_001168509.1	F: TCCTTGATCCTCACGGCCTA R: TTGGATGCTGCTTCCACGAT	194
p38	NM_001168508.1	F: GGTCTCACCACCTCAGTGTG R: GCTGTGGATGCCAGAACTCT	219
JNK1	NM_001168508.1	F: TGCCATCATGAGCAGAAGCA R: ATTCTGAAATGGCCGGCTGA	196
IL-10	NM_010548.2	F: TAAGGCTGGCCACACTTGAG R: CTCTGAGCTGCTGCAGGAAT	166
IL-6	NM_001314054.1	F: GGAGCCCACCAAGAACGATA R: TTGTGAAGTAGGGAAGGCCG	126
TNF-α	NM_001278601.1	F: CACAGAAAGCATGATCCGCG R: ACTGATGAGAGGGAGGCCAT	211
IL-1β	NM_008361.4	F: TCAGCACCTCACAAGCAGAG R: TTCTTGTGACCCTGAGCGAC	230

## Data Availability

The sequence datasets are available in GenBank and the accession number is OM679454-OM679997.

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
