# Peer review of "Arenga pinnata Resistant Starch Modulate Gut Microbiota and Ameliorate Intestinal Inflammation in Aged Mice"

_nutrients, 2022, doi:10.3390/nu14193931_

Round 1
Reviewer 1 Report
This paper is interesting and well-conceived. Briefly, it reports, compares, and discusses the effects of three Arenga pinnata resistant starch-based products (retrograded starch APRS, starch APS, and whole flour APF) on gut microbiota and their prebiotic activities with animal model (aged mice). In this context, the results show some relevant property-activity/function relationships (gut microbiota diversity index and composition, SCFAs production, intestinal barrier function, etc.), and well indicate potential beneficial effects of such products on human health.
I just suggest some minor revisions to improve the manuscript prior to its publication.
- Line 31: Write "100 trillion microbes" instead of 1x10E14 microbes;
- Line 65: "prebiotic" is the right word
- Lines 94-95: Remove the phrases "Mice were housed in..." & "The mice in all... feed and water" since already said previously (Line 90)
- Line 273: Add a comma (,) before respectively;
- Figure 6: Please indicate inside for each microscopic image the length scale;
- Line 369: Which physical and chemical properties might influence the alpha-diversity of microbial communities?
Author Response
Response to Reviewer 1 Comments
Point 1: This paper is interesting and well-conceived. Briefly, it reports, compares, and discusses the effects of three Arenga pinnata resistant starch-based products (retrograded starch APRS, starch APS, and whole flour APF) on gut microbiota and their prebiotic activities with animal model (aged mice). In this context, the results show some relevant property-activity/function relationships (gut microbiota diversity index and composition, SCFAs production, intestinal barrier function, etc.), and well indicate potential beneficial effects of such products on human health.
Response 1: Thank you for your attention on the manuscript. We appreciate your positive comments and sincerely accept your comments on this manuscript.
Point 2: I just suggest some minor revisions to improve the manuscript prior to its publication. Line 31: Write "100 trillion microbes" instead of 1x10E14 microbes;
Response 2: Thank you for your valuable comments. The information has been corrected in the manuscript.
Point 3: Line 65: "prebiotic" is the right word;
Response 3: Thank you for your valuable comments. The information has been corrected in the manuscript.
Point 4: Lines 94-95: Remove the phrases "Mice were housed in..." & "The mice in all... feed and water" since already said previously (Line 90);
Response 4: Thank you for your valuable comments. The information has been removed in the manuscript.
Point 5: Line 273: Add a comma (,) before respectively;
Response 5: Thank you for your valuable comments. A comma (,) has been added in the manuscript.
Point 6: Figure 6: Please indicate inside for each microscopic image the length scale;
Response 5: Thank you for your valuable comments. The length scale has been indicated in Figure 6 in the manuscript.
Point 7: Line 369: Which physical and chemical properties might influence the alpha-diversity of microbial communities?
Response 7: Thank you for your valuable comments. We speculated that there are some polyphenols in it that are causing the decrease in diversity of microbial communities. But Further experiments are needed to prove it. We would extract the polyphenols and evaluate their effects on the microbial communities. Thanks very much for your question again.
Finally, thank you again for your review and guidance!

Reviewer 2 Report
I congratulate the authors for this experimental study with aged mice evaluating the potential beneficial effect of Aregna pinnata products on human gut microbiota which is reflecting on the health status.
It is a well-designed study and all the methods followed have been described thoroughly which makes them reproducible by other investigators. However, statistical analysis should be expanded more, explaining all the tests used and how was normality assumed. In addition, sample size estimation should be added. Moreover, lines 200-214, should be omitted (probably the authors forgot to delete the instructions of the Nutrients Word Template needed for submission). The results are presented in a coherent manner and in an excellent sequence providing the maximum of the available information. The discussion is extensive enough with detailed reference to the available international literature.
To sum up, to my opinion this study is a good example of a well-designed experimental investigation and its results could be a springboard for further research on human beings.
Author Response
Response to Reviewer 2 Comments
Point 1: I congratulate the authors for this experimental study with aged mice evaluating the potential beneficial effect of Aregna pinnata products on human gut microbiota which is reflecting on the health status.
Response 1: Thank you for your attention on the manuscript. We appreciate your positive comments and sincerely accept your comments on this manuscript.
Point 2: It is a well-designed study and all the methods followed have been described thoroughly which makes them reproducible by other investigators. However, statistical analysis should be expanded more, explaining all the tests used and how was normality assumed. In addition, sample size estimation should be added.
Response 2: Thank you for your positive and valuable comments. The statistical analysis information has been added in the manuscript. The sample size was referred to our previous published literature and other published literatures (Ren, M.; Li, H.; Fu, Z.; Li, Q. Centenarian-Sourced Lactobacillus casei Combined with Dietary Fiber Complex Ameliorates Brain and Gut Function in Aged Mice. Nutrients 2022, 14, doi:10.3390/nu14020324; Sonoyama, K., Ogasawara, T., Goto, H., Yoshida, T., Takemura, N., Fujiwara, R., Yanagihara, T. (2010). Comparison of gut microbiota and allergic reactions in BALB/c mice fed different cultivars of rice. British Journal of Nutrition, 103(2), 218-226. https://doi.org/10.1017/s0007114509991589.; Zhang, Y., Chen, L., Hu, M., Kim, J. J., Lin, R., Xu, J., . Chen, S. (2020). Dietary type 2 resistant starch improves systemic inflammation and intestinal permeability by modulating microbiota and metabolites in aged mice on high-fat diet. Aging-Us, 12(10), 9173-9187. https://doi.org/10.18632/aging.103187.)。
Point 3: Moreover, lines 200-214, should be omitted (probably the authors forgot to delete the instructions of the Nutrients Word Template needed for submission).
Response 3: Thank you for your valuable comments. The information has been omitted in the manuscript.
Point 4: The results are presented in a coherent manner and in an excellent sequence providing the maximum of the available information. The discussion is extensive enough with detailed reference to the available international literature.
Response 4: Thank you for your positive comments.
Point 5: To sum up, to my opinion this study is a good example of a well-designed experimental investigation and its results could be a springboard for further research on human beings.
Response 4: Thank you for your positive comments.
Finally, thank you again for your review and guidance!
